# The Impact of the C-Terminal Region on the Interaction of Topoisomerase II Alpha with Mitotic Chromatin

**DOI:** 10.3390/ijms20051238

**Published:** 2019-03-12

**Authors:** Melissa Antoniou-Kourounioti, Michael L. Mimmack, Andrew C.G. Porter, Christine J. Farr

**Affiliations:** 1Department of Genetics, University of Cambridge, Downing St, Cambridge CB2 3EH, UK; melissa.ant.k@gmail.com (M.A.-K.); mlm20@medschl.cam.ac.uk (M.L.M.); 2School of Biological Sciences, University of East Anglia, Norwich Research Park, Norwich NR4 7TJ, UK; 3Metabolic Research Laboratories, Wellcome Trust-Medical Research Council Institute of Metabolic Science, University of Cambridge, Cambridge CB2 0QQ, UK; 4Centre for Haematology, Faculty of Medicine, Imperial College London, Hammersmith Hospital Campus, Du Cane Rd, London W12 0NN, UK; andy.porter@imperial.ac.uk

**Keywords:** topoisomerase II alpha, centromere, mitosis, chromosome, C-terminal domain, post-translational modification, SUMO, phosphorylation

## Abstract

Type II topoisomerase enzymes are essential for resolving DNA topology problems arising through various aspects of DNA metabolism. In vertebrates two isoforms are present, one of which (TOP2A) accumulates on chromatin during mitosis. Moreover, TOP2A targets the mitotic centromere during prophase, persisting there until anaphase onset. It is the catalytically-dispensable C-terminal domain of TOP2 that is crucial in determining this isoform-specific behaviour. In this study we show that, in addition to the recently identified chromatin tether domain, several other features of the alpha-C-Terminal Domain (CTD). influence the mitotic localisation of TOP2A. Lysine 1240 is a major SUMOylation target in cycling human cells and the efficiency of this modification appears to be influenced by T1244 and S1247 phosphorylation. Replacement of K1240 by arginine results in fewer cells displaying centromeric TOP2A accumulation during prometaphase-metaphase. The same phenotype is displayed by cells expressing TOP2A in which either of the mitotic phosphorylation sites S1213 or S1247 has been substituted by alanine. Conversely, constitutive modification of TOP2A by fusion to SUMO2 exerts the opposite effect. FRAP analysis of protein mobility indicates that post-translational modification of TOP2A can influence the enzyme’s residence time on mitotic chromatin, as well as its subcellular localisation.

## 1. Introduction

Type II Topoisomerases (TOP2) can pass one segment of DNA duplex through a transient double-strand break in a second segment in an ATP-dependent catalytic process. This allows the enzyme to resolve DNA topology issues arising during DNA replication and repair, transcription, chromatin remodelling and during chromosome compaction and segregation [1]. In eukaryotes the functional enzyme is formed through dimerization of TOP2 monomers, giving a structure that has three protein interfaces: the N-terminal ATPase gate (consisting of the ATPase and transducer domains); the DNA-binding gate (consisting of the TOPRIM domain, the Winged Helix Domain (with the active site tyrosine, Y805) and the Tower domain); and the C-gate (formed by the coiled-coil domain) [2,3,4]. The structure of these parts of the TOP2 dimer has been extensively investigated through X-ray crystallography [5,6,7]. However, outside this catalytic core, the enzyme has a less conserved, catalytically-dispensable subdomain, the C-Terminal Domain (CTD). The eukaryotic CTD has not been crystallised and appears to be structurally disordered.

In vertebrates two isoforms of TOP2 exist, alpha and beta, encoded by separate genes [8,9,10]. Many of the key differences between the two isoforms are determined by their distinct CTDs [11,12]. The start of the CTD in human TOP2A has been placed variously at residue 1173 up to amino acid 1244, depending on whether sequence alignment or partial proteolysis, has been used to define it [11,12,13]. This region of the protein is required for its nuclear localisation [14,15], has been shown to modulate the enzyme’s activity [12,16,17] and to be responsible for the preference of the alpha isoform for relaxing positively-supercoiled DNA [18,19,20]. TOP2 (TOP2A in vertebrates) associates dynamically with chromosome arms throughout mitosis and is found concentrated at centromeres from prophase through until anaphase onset [21,22,23,24,25,26,27]. In vertebrates, the alpha-CTD is responsible for TOP2A targeting mitotic chromatin [11,28] and for facilitating the recruitment of specific proteins to the mitotic centromere [29,30]. 

The alpha-CTD has several clusters of positively-charged amino acids (lysine and arginine residues), which may influence the protein’s association with negatively-charged DNA [31]. One cluster, near the C-terminus of human TOP2A (amino acids 1490–1492), is part of a functional bipartite nuclear localisation signal (NLS) [14,15,32], while lysines within the extreme C-terminal 31 amino acids (1501–1531) form part of the more recently identified chromatin tether domain (ChT), that is essential for TOP2A to interact robustly with chromosomes in mitosis [28]. 

The CTD is rich in residues subject to modification, with multiple sites of phosphorylation, SUMOylation, ubiquitination and acetylation identified [13,33] (PhosphositePlus database url: https://www.phosphosite.org). This may partly reflect the disordered and accessible nature of this protein domain [34,35,36]. TOP2 post-translational modifications have been found to influence its catalytic activity, protein stability and subcellular distribution [37]. For example, work using *Xenopus* egg extracts (XEE) has shown that Top2a is a major SUMOylation target during mitosis, with the modified protein concentrated at the centromere [38,39,40]. Subsequently SUMOylation of specific acceptor sites within the *Xenopus* Top2a CTD was shown to influence Claspin and Haspin Kinase recruitment to the mitotic vertebrate chromosome [29,30]. Further evidence, for CTD SUMOylation being involved in the recruitment of Haspin Kinase and of Aurora B Kinase, has come from studies in *S.cerevisiae* [41]. Meanwhile work in human cell lines and in transgenic mice, has shown that perturbations in SUMO ligase activity and disruption of TOP2A SUMOylation, reduces chromosome segregation fidelity [27,42]. However, the molecular mechanisms underlying these effects remain unknown. Given the large number of modifiable sites and their potential for cross-talk, the dynamic combination of modifications present on different pools of TOP2 molecules through the cell cycle is likely to be highly complex and its biological significance is, as yet, largely unexplored. 

Here we show that post-translational modification of specific residues within the CTD influences the behaviour of human TOP2A in mitosis: SUMOylation and phosphorylation impact on the dynamic exchange of TOP2A on mitotic chromatin and on the efficiency with which the protein is maintained at the centromere as cells progress towards anaphase onset.

## 2. Results

### 2.1. The Impact of Internal Deletion of the CTD on Localisation of TOP2A to Mitotic Chromatin

Previous work has shown that the CTD of human TOP2A (residues 1173–1531) is required for efficient localisation to mitotic chromatin [11]. Subsequently Clarke and colleagues demonstrated that the most distal 31 amino acids, as well as encompassing the main nuclear localisation signal (NLS), are crucial for localisation to mitotic chromatin. They designated this component the chromatin tether (ChT). However, they also concluded that, while important, the ChT does not function in isolation and that other parts of the CTD contribute to the protein’s robust localisation to mitotic chromosomes [28].

Stable human cell lines were established expressing internally deleted forms of human TOP2A (Figure 1a). The parent cell line was a HT1080 conditional null mutant, HTETOP. In these cells both endogenous *TOP2A* alleles have been disrupted and expression of an exogenous wild type (WT) *TOP2A* cDNA is controlled by a Tet transactivator (tTA) [43]. This allows the wild type transgene’s expression to be repressed by doxycycline (dox), with TOP2A protein levels falling to <1% over 3–4 days, with lethal consequences [43,44,45]. The parent cell line was transfected with expression constructs encoding several, internally deleted, forms of TOP2A tagged at the N-terminus with the Flag epitope. In each case the constitutively-expressed mutant retained the terminal amino acids 1447–1531, which encompass the main NLS [14,32] and the ChT domain [28]. Stable transfectants were established in the absence of doxycycline (i.e., expressing the untagged full length TOP2A protein) and the presence of the Flag-tagged protein was confirmed by immunoblotting (Figure 1b). The ability of mutant protein to rescue established clones from dox-induced lethality was then assessed. 

The most extensively deleted form of TOP2A tested (FTΔ2), in which residues 1174–1446 had been removed, was found not to complement loss of full length TOP2A in HTETOP cells (Figure 1b, Table 1). Immunofluorescence revealed that nuclear localisation and mitotic chromatin association of this variant were both severely compromised, despite it retaining the major NLS and ChT domain (Figure 1c). This mislocalisation was observed irrespective of whether cells were grown in the absence, or presence (48 h), of doxycycline. This suggests that deletion of amino acids extending as far proximal as residue 1174 disrupts the overall function of the enzyme, possibly by impinging on the structure of the C-gate formed by the coiled coil region [13].

Stable cell lines were generated expressing variants with smaller internal deletions, spanning residues 1212–1446 (FTΔ5) and 1321–1446 (FTΔ3). Both variants complement loss of full length TOP2A, showing that the mutant homodimers are catalytically active in vivo. This is consistent with previous reports of stable HTETOP transfectants rescued from dox-lethality by expression of a TOP2A variant truncated at residue 1201, while retaining the main NLS (residues 1454 to 1497) [17]. It is also consistent with truncation of the *S.cerevisiae* top2 protein at 1209 (equivalent to reside 1217 in human TOP2A). This truncated yeast protein has been reported as normal, when examined using electron microscopy and is active in vivo, complementing top2 loss [46,47]. When FTΔ3 and FTΔ5 cells were examined using immunofluorescence both variants displayed normal nuclear localisation (Figure 1c). In mitosis, the Δ3 form was found to accumulate on chromatin, displaying an axial distribution and punctate staining at centromeres comparable to full length TOP2A, irrespective of the presence/absence of doxycycline. In FTΔ5 cells localisation to mitotic chromatin was disrupted: in the presence of doxycycline the variant was diffusely associated with chromatin, while in the presence of full length TOP2A (-dox) the Flag-tagged variant was largely cytosolic. This suggests that deletion of residues 1212-1446 disrupts the targeting of TOP2A to mitotic chromatin, a phenotype compounded by the presence of full length TOP2A. 

Overall these results suggest that multiple sequences within the CTD contribute to the robust mitotic localisation of TOP2A, including residues within the proximal part of the domain (amino acids 1212–1320) (Table 1). 

### 2.2. Candidate SUMOylation Sites within the TOP2A C-Terminal Region (CTR)

Mass spectrometry studies have reported candidate SUMO acceptor sites throughout TOP2A, including many in the CTD (PhosphositePlus database: https://www.phosphosite.org/). Using this information, together with SUMO site prediction software (SUMOsp 2.0 [48,49] and SUMOplot (SUMOplot Analysis Program, https://www.abgent.com/tools)) eleven candidate acceptor lysines within the C-terminal region of TOP2A were selected and replaced by arginine and the impact of these substitutions on TOP2A SUMOylation assayed through Hek-293T transient transfection and immunoblotting. Since SUMOylation is a highly dynamic process and, like most SUMO targets, only a minor TOP2A fraction is SUMOylated at any moment in time [38,39,42,50,51,52] steps were taken to enhance the sensitivity of the assay: (i) the ectopically expressed proteins were tagged: SUMO peptides with either Haemagglutinin (HA) or Green Fluorescent protein (GFP) and TOP2A with Flag; (ii) a general inhibitor of cysteine proteases, N-ethylmaleimide and (iii) Q90P variants of SUMO2 and SUMO3 were used [53], to slow down de-conjugation; and (iv) since TOP2A is a relatively large protein (~170 KDa) the assay focussed on the C-terminal third (amino acids 971–1531). 

Plasmid DNAs encoding epitope-tagged SUMO1, SUMO2 or SUMO3 were co-transfected with DNA encoding the Flag-tagged C-terminal 560 residues of TOP2A (referred to here as the CTR to distinguish it from the CTD defined by proteolysis and sequence conservation) (Figure 2a). Immunoblotting, using an anti-Flag antibody, revealed the TOP2A CTR (~100 KDa), together with a ladder of higher molecular weight (HMW) products (Figure 2b,c). This ladder was detected when the CTR expression construct was co-transfected with either SUMO1, 2 or 3 but differed slightly depending on whether the SUMO moiety (~90 amino acids) had been tagged with the 9 amino acid HA epitope or with Green Fluorescent protein (GFP, 238 amino acids) (Figure 2b–d). In the absence of any ectopically expressed SUMO a weak ladder of HMW forms of the CTR was detected, reflecting low level conjugation involving the endogenous pool of SUMO peptides (Figure 2c). The HMW ladder detected with co-expressed HA:SUMO has a similar pattern of regularly spaced but more intense, bands consistent with the attachment of multiples of the SUMO moiety [54]. For GFP-tagged SUMO the ladder is less regular, with bands varying in strength. This may be due to GFP (which is more than twice the size of the SUMO peptide) interfering, to some extent, with SUMOylation, as well as conjugation of endogenous SUMO peptides (15–20 KDa) occurring alongside GFP:SUMO (~40 KDa). Bands of ~140 and 180 KDa are apparent, consistent with attachment of one and two GFP:SUMO units respectively.

To demonstrate directly that the HMW bands represent SUMOylated forms of the TOP2A CTR, GFP immunoprecipitations were performed on cells co-transfected with Flag-tagged CTR and either eGFP alone or eGFP-tagged SUMO3. Western blotting of the immunoprecipitated samples with anti-Flag antibody established that the TOP2A CTR is SUMOylated during transient transfection of HEK-293T cells (Figure 2d). As discussed previously, the presence of some weaker intensity bands within the Flag:CTR ladder may reflect molecules modified by a mix of endogenous SUMO and ectopic GFP:SUMO moieties. In addition, since the CTR retains dimerization capacity, some molecules lacking attached SUMO/GFP:SUMO may be immunoprecipitated as a result of heterodimerisation between modified and unmodified substrate molecules.

Fluorescence immunoblotting was used to quantitate the ladder of HMW forms relative to the unmodified ~100 KDa. Flag:CTR. Levels were estimated in the absence (endogenous SUMO only) and presence of various forms of HA-tagged SUMO. In the presence of ectopically-expressed SUMO1 or SUMO2 the ratio of the HMW ladder relative to the unmodified Flag:CTR was ~4–6-fold greater than that detected for the endogenous SUMO-only control (Figure 2e).

Transient expression of HA-tagged SUMO2 Q90P was used to determine whether mutating putative SUMO acceptor sites impacts on modification of the CTR. When the CTR domain contained arginine substitutions at all eleven selected lysines (11× KR) its SUMOylation was reduced to ~13% that of the wild type CTR (Figure 2c,f). A substantial portion of this effect could be accounted for by mutating K1240, either alone (36% of WT) or in combination with the three adjacent lysines, K1228, K1264 and K1267 (17–31% of WT) (Figure 2f). While sites other than K1240 were not tested individually, these data indicate that K1240 acts as a major TOP2A SUMO acceptor site when the CTR is transiently overexpressed in cultured human cells.

### 2.3. SUMO-Phospho Cross Talk

The peptide sequence flanking K1240 includes downstream threonine and serine residues (I**K**ENENTEGSP) (Figure 2a). This is of interest because an extended SUMO consensus sequence, ϕKxExxSP, has been identified in several proteins that are substrates for both SUMO conjugation and proline-directed kinases [55,56]. Moreover, TOP2A peptides simultaneously SUMOylated at K1240 and phosphorylated at S1247 have been reported in mass spectrometry screens [36,57,58]. Both T1244 and S1247 have been identified as phosphorylation sites, with S1247 hyperphosphorylated in G2/M [59,60] (PhosphositePlus database: https://www.phosphosite.org/). Therefore, the effect on TOP2A CTR SUMOylation of mutating T1244 and S1247, either to non-phosphorylatible alanine or to the phosphomimetic aspartic acid, was assessed. Mutation of either site to alanine (singly or together) significantly reduced the level of the Flag-tagged HMW ladder compared to that of the unmutated CTR (Figure 2g). Conversely, introduction of aspartic acid at both sites (giving the double mutant T1244D and S1247D) enhanced the CTR SUMOylated ladder significantly (Figure 2g). This suggests that the phosphorylation status of residues in the vicinity of K1240 has the potential to influence SUMO modification.

### 2.4. Mutation of TOP2A SUMO Acceptor Sites and Cell Cycle Progression

To investigate possible effects of TOP2A SUMOylation on mitosis, human cell lines stably expressing full length TOP2A variants were generated in the HTETOP cell background. The modifications included: (i) K/R substitution of candidate SUMO acceptor sites; (ii) constitutive, in-frame, SUMO fusions; and (iii) S/A or S/D mutation of sites hyperphosphorylated in G2/M. In each, full length TOP2A protein was N-terminally tagged, either with a 3× Flag epitope or GFP (summarised in Figure 3a). Quantitative immunoblotting was used to identify clones expressing epitope-tagged TOP2A protein at levels comparable with that of the untagged TOP2A in the parental HTETOP cells (cells grown without doxycycline) (Figure 3b). All variants tested complemented the lethal effects of doxycycline exposure (i.e., loss of wild type TOP2A). 

Mitotic progression was examined in cells stably expressing, as their only TOP2A form, either Flag-tagged full length wild type protein, protein with lysine to arginine mutations at the eleven candidate acceptor sites examined in the transient transfection assay (variant FT-11xKR) or a Flag:SUMO2:TOP2A fusion protein (FST). Examination of fixed cells from asynchronously growing populations grown in the presence of doxycycline (to suppress the wild type transgene) revealed no marked difference in mitotic indices (while the Flag transfectants tested showed a slight decrease in mitotic index compared to parental HTETOP cells, the cell lines expressing the variant forms of Flag:TOP2A were consistent with the control Flag:TOP2A WT-expressing cells). Furthermore, there were no statistically significant differences in the distribution of cells across M-phase or in the numbers of abnormal nuclear phenotypes (specifically, apoptotic nuclei, micronuclei or teardrop-shaped nuclei) (Figure 3c,d). 

The fidelity of mitotic chromosome segregation was assessed in anaphase cells (Figure 3e). Included in this analysis were cells known to display high levels of chromosome mis-segregation: the parental HTETOP cell line grown in doxycycline for 48 h (to lower TOP2A protein levels) and FT-K662R. K662 lies within the DNA binding domain and in vitro studies have shown that substitution of this amino acid reduces the enzyme’s catalytic activity [40,61]. This mutant is viable in human cells in the absence of wild type TOP2A, despite displaying less efficient chromosome compaction and a longer population doubling time [45]. When TOP2A is depleted ~80% of anaphase cells display chromosome mis-segregation (Figure 3e). A less severe defect (~40% mis-segregation) was seen in cells expressing TOP2A K662R (Figure 3e). In contrast, cell lines expressing TOP2A with eleven K/R substitutions within the C-terminal region or the Flag-SUMO2-TOP2A (FST) fusion protein, displayed robust chromosome segregation, comparable with the parental HTETOP cell line grown without doxycycline (Figure 3e).

Mitotic progression was examined in cells stably expressing, as their only TOP2A form, either Flag-tagged full length wild type protein, protein with lysine to arginine mutations at the eleven candidate acceptor sites examined in the transient transfection assay (variant FT-11xKR) or a Flag-SUMO2-TOP2A fusion protein (FST). Examination of fixed cells from asynchronously growing populations grown in the presence of doxycycline (to suppress the wild type transgene) revealed no marked difference in mitotic indices. [While the Flag transfectants tested showed a slight decrease in mitotic index compared to parental HTETOP cells, the cell lines expressing the variant forms of Flag:TOP2A were consistent with the control Flag:TOP2A(WT)-expressing cells]. Furthermore, there were no statistically significant differences in the distribution of cells across M-phase or in the numbers of abnormal nuclear phenotypes (specifically, apoptotic nuclei, micronuclei or teardrop-shaped nuclei) (Figure 3c,d). 

The fidelity of mitotic chromosome segregation was assessed in anaphase cells (Figure 3e). Included in this analysis were cells known to display high levels of chromosome mis-segregation: the parental HTETOP cell line grown in doxycycline for 48 h (to lower TOP2A protein levels) and FT-K662R. K662 lies within the DNA binding domain and in vitro studies have shown that substitution of this amino acid reduces the enzyme’s catalytic activity [40,61]. This mutant is viable in human cells in the absence of wild type TOP2A, despite displaying less efficient chromosome compaction and a longer population doubling time [45]. When TOP2A is depleted ~80% of anaphase cells display chromosome mis-segregation (Figure 3e). A less severe defect (~40% mis-segregation) was seen in cells expressing TOP2A K662R (Figure 3e). In contrast, cell lines expressing TOP2A with eleven K/R substitutions within the C-terminal region or the Flag:SUMO2:TOP2A (FST) fusion protein, displayed robust chromosome segregation, comparable with the parental HTETOP cell line grown without doxycycline (Figure 3e).

### 2.5. Factors Influencing TOP2A Localisation in Mitosis

#### 2.5.1. SUMOylation

During mitosis TOP2A accumulates on mitotic chromosomes, displaying a characteristic axial distribution along the arms. In addition, strong punctate TOP2A signals are detectable at centromeres during early mitosis (prophase, prometaphase and metaphase) (Figure 4a–c). The immunofluorescence pattern in prometaphase/metaphase cells expressing Flag-tagged wild type TOP2A was compared to that in cells with K/R substitutions within the C- terminal region (FT-11xKR, FT-4xKR and FT-K1240R), in the catalytic domain (FT-K662R) and in cells expressing a constitutive SUMO2:TOP2A fusion protein (FST) (Figure 4d). Mutation of candidate lysines within the C-terminal region resulted in a significant decrease in the fraction of prometaphase/metaphase cells displaying TOP2A accumulation at centromeres relative to chromosome arms. The K662R substitution also decreased the number of cells displaying punctate centromere localisation relative to chromosome arms. Conversely, constitutive SUMOylation significantly increased the fraction of cells showing centromeric accumulation (Figure 4d). These effects were seen irrespective of the N-terminal epitope tag used (Flag or GFP) and, for the constitutively SUMOylated protein, independent of whether the SUMO2 peptide was attached to the N- or C-terminus of TOP2A (FST and GT:SUMO2 respectively) (Figure 4d,e). 

#### 2.5.2. Phosphorylation

The impact on TOP2A mitotic localisation of mutating two serines in the CTD, known to be hyperphosphorylated during G2/M, was examined. Transient transfection assays had revealed that S1247 phosphorylation may influence SUMOylation of nearby K1240, while other approaches have shown that a population of TOP2A molecules phosphorylated on S1213 localises specifically to the centromere during mitosis [62]. Substitution of serine by alanine at either site significantly decreased the number of cells displaying centromeric accumulation of TOP2A in prometaphase and metaphase (Figure 4e). 

Both S1213 and S1247 conform to the proline-dependent phosphorylation site characteristic of CDK1/Cyclin B [63,64]. To further examine the impact of CDK1 phosphorylation on the localisation of TOP2A during M phase, cells expressing WT or constitutively-SUMOylated TOP2A were treated overnight with the reversible CDK1 inhibitor RO-3306, leading to arrest at the G2/M border [65,66]. One hour after release into drug-free medium, cells were prepared for immunofluorescence and the mitotic localisation of TOP2A assessed (Figure 4f). In both cell lines, pre-treatment with RO-3306 resulted in a significant decrease in the number of cells showing centromeric accumulation of TOP2A at prometaphase/metaphase. This suggests that disruption of CDK1 activity levels and prolonged arrest at the G2/M boundary, impacts on centromeric localisation of TOP2A in the following M phase. To determine whether the CTD mutations examined exert an effect on centromeric accumulation at the onset of mitosis TOP2A localisation was assessed in prophase cells. The majority of prophase cells displayed centromeric TOP2A accumulation, irrespective of whether they expressed WT or phospho-mutant TOP2A (Figure 4g). The same robust accumulation at the prophase centromere was also seen in cells expressing the K1240R variant (Figure 4g). This suggests that these mutations do not affect the ability of TOP2A to localise to the centromere at mitotic entry but instead impact on the efficiency with which it is maintained there as cells progress towards anaphase onset. 

### 2.6. Factors Influencing the Dynamic Exchange of TOP2A on Mitotic Chromatin

Fluorescence Recovery after Photobleaching (FRAP) has revealed that TOP2A exchanges dynamically between chromatin and the cytosol during mitosis, with a half-life of seconds [26,28,67]. To determine whether the TOP2A substitutions studied here impact on the way that the protein interacts with mitotic chromatin, FRAP was used on live cells expressing TOP2A N-terminally tagged with GFP (Figure 3a). These cell lines were grown continuously in doxycycline, allowing FRAP to be performed in the absence of untagged TOP2A. A cell line expressing GFP-tagged histone H2B (H2B:GFP) was used as a negative control, since canonical histone proteins are immobile during mitosis. During FRAP a small area of the chromosome arm was bleached using high intensity light, with the fluorescence recovery time taken as an indicator of the protein’s dynamics: a faster recovery time suggests a shorter residence time on chromatin, while a slow recovery suggests a long residence time (Figure 5a) [68]. 

Recovery is dependent on two factors: (i) protein diffusion and (ii) chemical interactions. The free protein within the region of interest (r.o.i.) will diffuse readily outside the r.o.i., allowing unbleached protein to enter. Protein that is attached to other structures (such as mitotic chromatin) will be slower to exit the r.o.i. Therefore, this fraction of the protein will occupy space in the r.o.i. for longer, not allowing free unbleached molecules to enter. Recovery is fitted to a bi-exponential equation and two t_1/2_ values (times to half-maximum) calculated: a fast t_1/2_ (mainly dependent on diffusion) and a slow t_1/2_ (dependent on the chemical interaction). The recovery curves of different proteins/variants can be compared directly to give a goodness-of-fit value (R^2^). 

We examined the impact on the mitotic chromatin dynamics of TOP2A of two K/R substitutions: K1240R and K662R. The catalytically-compromised GFP:TOP2A K662R (GT-K662R) displays a slower recovery than unmutated TOP2A (GFP:TOP2A) (Figure 5b). Both the fast and slow t_1/2_ values are greater in cells expressing the K662R variant, with the recovery curve showing a significant deviation from that displayed by the unmutated protein (Table 1). In cells expressing GFP:TOP2A K1240R (GT-K1240R), while the fast t_1/2_ was less than for GFP:TOP2A, the slow t_1/2_ and overall recovery curve did not differ significantly from wild type (Figure 5c, Table 2). 

In cells ectopically expressing GFP:TOP2A with a SUMO2 peptide attached to the C-terminus (GT:SUMO2) both the fast and slow t_1/2_ values were shorter, with the shape of the recovery curve differing significantly from that of GFP:TOP2A (Figure 5b, Table 2). We also examined the protein’s dynamics when the SUMO2 peptide was placed at the TOP2A N-terminus (G:SUMO2:T). Although the extent of the effect was less dramatic than when the SUMO peptide was located at the TOP2A C-terminus nevertheless, as for the GT:SUMO2 fusion protein, both the fast and slow t_1/2_ values were shorter than for GFP:TOP2A (Figure 5b, Table 2).

The impact on TOP2A dynamics of mutating two serines within the CTD, known to be hyper-phosphorylated in M phase, was also examined. The recovery time of the S1247A variant was indistinguishable from that of the wild type protein. In contrast substituting serine 1213 for alanine dramatically slowed its recovery and was the only variant tested that did not reach full recovery by 4 min post-bleach (Figure 5c, Table 2). To examine the role of S1213 on TOP2A dynamics in mitosis further, the serine was substituted for a phosphomimetic residue, aspartic acid, with the recovery of both variants followed over an extended time period (6 min). The post-bleach time course again revealed that the S1213A substitution produces a slow recovery, with an overall recovery curve significantly different to that of the unmutated protein. It takes >300 s for the S1213A protein to recover to levels similar to GFP:TOP2A. While the S1213D variant exhibited a shorter slow t_1/2_ value than the WT protein, overall its recovery curve is not significantly different from wild type, suggesting that the presence of a phosphomimetic residue at position 1213 does not increase the protein’s rate of recovery above that of the wild type (Figure 5d, Table 3).

## 3. Discussion

Mitotic chromosomes are dynamic structures whose shape and make-up change as cells proceed through M phase [70]. Condensin II loads onto mitotic chromatin during prophase, while condensin I, which is cytoplasmic during interphase, only gains access after nuclear envelope breakdown (NEBD) [71]. Cohesin is extensively removed from arms during prophase [72]. Proteins of the Chromosome Passenger Complex, such as INCENP, localise to chromosome arms during prophase but become progressively concentrated at the centromere during prometaphase/metaphase [73]. Histone H3 modified by serine 10 phosphorylation appears in late S/G2 at pericentromeric regions, before spreading along chromosome arms as mitosis proceeds, where it persists until late anaphase/telophase [74,75]. In contrast H3 threonine 3 phosphorylation only appears during prophase, concentrating at centromeres in prometaphase/metaphase and is lost at anaphase [76,77]. TOP2 (TOP2A in vertebrates) localises to chromosome arms from prophase up until telophase and accumulates at centromeres during prophase, prometaphase and metaphase [21,22,23,24,25,26,27]. In this study, the impact on the behaviour of TOP2A during mitosis of the CTD and its post-translational modification, has been explored.

We report that human TOP2A from which amino acids 1212–1446 have been removed, while complementing loss of the wild type protein, localises less efficiently to mitotic chromatin. This deleted form retains the protein’s main NLS and the ChT domain, essential for it to target mitotic chromatin, suggesting additional contributions from other residues within the CTD [28]. Given reports that mitotic localisation of vertebrate TOP2A is influenced by SUMOylation [27,38,39,40,42] we used a transient transfection assay to test the significance in vivo of candidate acceptor lysines within the CTD. The findings suggest that lysine 1240 is a major SUMOylation target in cycling human cells. Moreover, we show that SUMOylation efficiency can be influenced by phosphorylation of nearby threonine 1244 and serine 1247. The importance of K1240 as a SUMO acceptor is in agreement with the key SUMO sites identified within the *Xenopus* Top2a CTD [29] and in *S.cerevisiae* top2 [51,78]. Moreover, the evidence for K1240 being a phosphorylation-dependent SUMOylation site is supported by the identification of co-modified peptides in mass spectrometry screens [36,57,58], together with the finding that such SUMO-phospho cross talk is enriched for cyclin-dependent kinases (both S1213 and S1247 lie within CDK1 recognition sites) and for disordered and flexible protein regions (like the CTD) [36]. Nevertheless, substitution of all eleven selected lysines within the human TOP2A C-terminal region did not eliminate its SUMOylation completely in this assay. This may be explained by the fact that large scale proteomics has identified ~100 SUMO2/3 sites in human TOP2A. These are distributed across the protein, mapping not only within the CTD but also to the N-terminal ATPase and DNA binding domains [36,40,57,58,79,80,81,82,83,84]. 

Unlike the situation where TOP2A protein is depleted from cells (HTETOP cells treated with doxycycline) or where the enzyme’s catalytic activity is compromised (TOP2A K662R), the presence of eleven K to R substitutions within the C-terminal region had no obvious impact on mitotic progression or on the fidelity of chromosome segregation. This is consistent with the findings in *S.cerevisiae* where mutation of multiple predicted acceptor lysines within the C-terminal domain, while substantially reducing Top2 SUMOylation, did not result in any overt growth or chromosome segregation defects [51,78]. These observations are in contrast to studies in human cells and transgenic mice in which TOP2A SUMOylation was disrupted through knock-down of SUMO ligase activity: in these studies, the fidelity of chromosome segregation was reduced [27,42]. This suggests that either SUMOylation of additional sites within TOP2A and/or of additional proteins contributes to reduced chromosome segregation fidelity following SUMO ligase perturbation. 

Both the K662R and the K/R CTD substitutions analysed (K1240R alone or together with the other selected K/R substitutions within the CTD) were found to influence TOP2A mitotic localisation, resulting in fewer cells displaying centromeric accumulation during prometaphase-metaphase. The same localisation phenotype was displayed by prometaphase-metaphase cells expressing TOP2A in which the serine residue at either 1213 or 1247 (sites known to be hyperphosphorylated during mitosis) had been substituted by alanine. Conversely, a fusion protein in which a SUMO2 peptide had been covalently attached to TOP2A, exerted the opposite effect, increasing the proportion of cells displaying prometa-/metaphase centromeric TOP2A accumulation. This enhanced centromere accumulation phenotype was seen irrespective of the N-terminal tag employed (GFP or Flag) or of whether SUMO2 was placed at the N- or C-terminus of TOP2A and is in agreement with the findings of Dawlaty and colleagues [42].

Overall these results suggest that the retention of TOP2A at centromeres as M phase proceeds is influenced by the protein’s phosphorylation and SUMOylation status. To better characterise the underlying molecular mechanisms(s) FRAP was used to examine the mobility of these various forms of TOP2A. This revealed that fusing a SUMO2 peptide to the TOP2A terminus, as well as enhancing its accumulation at the mitotic centromere, also results in an increase in the enzyme’s mobility on mitotic chromatin. While a SUMO fusion protein is an artificial construct, that does not recapitulate the normally dynamic nature of this modification, it does provide one tool with which to explore the possible effects of TOP2A SUMOylation. Its shorter residence time on mitotic chromatin suggests a less robust interaction. This might be accounted for by the fusion protein’s lower isoelectric point (pI) disrupting its interaction with DNA or through changes in SUMO-mediated interactions with other chromosomal proteins. The increased mobility of SUMOylated TOP2A is consistent with an earlier report, using *Xenopus* egg extracts, that SUMO conjugation decreases the affinity of a small fraction of top2a for mitotic chromatin [38]. 

Given the large number of potential SUMO acceptor lysines in TOP2A the impact of SUMOylation in vivo is likely to be complex, with modification of individual sites exerting different effects [29]. In the current study, no decrease in TOP2A mobility was detected when one of its major SUMO acceptor sites, K1240, was substituted for arginine, despite this mutation affecting localisation at the centromere. The absence of any significant effect on mobility may be accounted for by the fact that the K1240R variant remains permissive to SUMOylation at many other acceptor sites. Intriguingly, however, the K662R substitution did result in TOP2A having significantly slower dynamics but whether this effect is mediated by a change in the mutated protein’s SUMO status remains unknown. This residue is highly conserved within the catalytic domain and directly involved in binding the G-segment DNA [4]. It lies in a key position in the DNA gate domain and in vitro decatenation assays have revealed that mutation of this lysine substantially reduces the enzyme’s activity [40,61]. In *S.cerevisiae*, a top2 mutant in which the equivalent residue is mutated to alanine (K651A) is non-viable [61], while human cultured cells expressing only the K662R substituted form are viable but their growth is compromised [45]. The longer residence time of TOP2A K662R on mitotic chromatin suggests that disruption of this variant’s activity may involve a delay in the catalytic cycle that results in the enzyme remaining associated with mitotic chromatin for longer than normal. This reduced mobility correlates with less efficient TOP2A localisation at the centromere during prometa-/metaphase. 

As well as SUMOylation, we also examined the impact on mobility of mutating two serines hyperphosphorylated in mitosis: S1213 and S1247. Phosphorylation of TOP2A on S1213 is M-phase-specific and this pool of TOP2A molecules has been reported to concentrate at centromeres during metaphase [62,63]. Recently peptides co-modified by phosphorylation at S1213 and SUMOylation at K1204 have been identified by mass spectrometry, raising the possibility that this phospho site may also influence the efficiency of SUMO-modification at specific sites within the CTD [36]. Here we report that, in cells expressing TOP2A S1213A as their only form of TOP2A, the protein has a longer residence time on the mitotic chromosome than wild type (or TOP2A S1213D), as well as exhibiting less efficient accumulation at centromeres. In contrast, the S1247A substitution did not have any obvious effect on the protein’s mobility, despite reducing its accumulation at the centromere.

Together these findings suggest that SUMOylation and phosphorylation of specific residues within the TOP2A CTD can regulate the enzyme’s residence time on mitotic chromatin and, through this, influence the protein’s subcellular localisation, with a faster exchange between molecules in the chromatin and cytosolic pools apparently facilitating centromere targeting. However, our data also suggest that residence time is not the only factor that can influence the ability of TOP2A to accumulate at the centromere, since two mutations, K1240R and S1247A, disrupt centromeric localisation without having any obvious impact on protein mobility. Work reported here and elsewhere, suggest that K1240 is an important SUMO acceptor site in TOP2A, with this modification being influenced by phosphoS1247. Lysine 1240 in human TOP2A is a highly conserved residue that lies within an optimal SUMO consensus motif in a wide range of eukaryotes [29,51,78,85]. One possibility is that SUMOylation at K1240 influences the interaction of TOP2A with centromere proteins that have SUMO interacting motifs (SIM), rather than with mitotic chromatin more generally [29,42,85,86]. Indeed, work from others has reported that SUMOylation of top2a in *Xenopus* at a limited number of sites within the CTD (including the lysine analogous to human K1240) is required for recruitment of Claspin and Haspin Kinase to the inner centromere domain during M-phase [29]. 

The work described here has identified specific residues within the human TOP2A CTD (K1240, S1213 and S1247) whose post-translational modification influence the protein’s localisation at the mitotic centromere. The molecular mechanism(s) underlying this regulation remain to be identified. Since it has not been possible to crystallise the CTD of eukaryotic TOP2 we can only speculate on the possible impact of these post-translational modifications on the protein’s overall tertiary structure. Nevertheless, both SUMOylation and phosphorylation, of mitotic TOP2A will change the protein’s pI, possibly disrupting its interaction with DNA and promoting faster exchange between molecules in the chromatin and cytosolic pools. In parallel with this, TOP2A SUMOylation may increase its residence time at the centromere through specific, SUMO-directed, protein-protein interactions.

## 4. Materials and Methods

### 4.1. Antibodies

Primary antibodies used for immunoblotting were: anti-human TOP2A (mouse, MBL, 1:5000); anti-GFP (mouse, Roche, Basel, Switzerland, 1:2000); anti-HSP70 (mouse, Santa Cruz, Dallas, TA, USA, 1:4000); anti-α-tubulin (mouse, Abcam, Cambridge, UK, 1:10,000); anti-Flag M2 (mouse, Sigma, St. Louis, MO, USA, 1:4000); anti-Cyclin B1 (mouse, BD, Franklin Lakes, NJ, USA, 1:500). Secondary antibodies were: IRDye 800CW goat anti-mouse IgG (H + L) (LI-COR, 1:7000); poly-HRP goat anti-mouse (ThermoFisher Scientific, Waltham, MA, USA, 1:15,000). For indirect immunofluorescence the primary antibodies used were: anti-human topoisomerase 2α (mouse, MBL, Woburn, MA, USA, 1:500); anti-human CENP-C (rabbit, 1:1000); and anti-Flag M2 (mouse, Sigma, 1:1000). Secondary antibodies were: rabbit anti-mouse FITC (Dako, Glostrup, Denmark, 1:200); Donkey anti-mouse Alexa-Fluor 488-conjugated (ThermoFisher, 1:2000); Donkey anti-rabbit Alexa Fluor 647-conjugated (ThermoFisher, 1:2000). 

### 4.2. Molecular Biology

TOP2A expression constructs used and/or generated as part of this study are detailed in Appendix A. All N-terminal GFP:TOP2A plasmid constructs used in this study were based on pGFP:TOP2A [43]. This plasmid carries a puromycin-resistance cassette. To generate expression constructs encoding TOP2A tagged at its N-terminus with the Flag epitope, the pcDNA3 vector (Invitrogen) was modified to carry a puromycin-resistance cassette (from pPUR, Clontech, Mountain View, CA, USA) in place of the SV_2_neo^r^ cassette (designated pcDNA3-puro). Three copies of the Flag epitope were cloned into the multi-cloning site, using HindIII and EcoRI. Finally, the open reading frame (ORF) of human TOP2A was introduced as an EcoRV-NotI fragment [45]. The QuikChange site-directed mutagenesis Kit was used to generate specific point mutations according to the manufacturer’s instructions (Stratagene, San Diego, CA, USA). Mutagenesis was performed on appropriate sub-fragments and sequencing used to confirm the presence of desired mutations, before and after reintroduction of the mutated sub clones back into the TOP2A expression construct. For transient transfection assays, HA-tagged and GFP-tagged SUMO expression constructs encoding SUMO mature forms (SUMO-GG) were used [87,88]. Q90P variants of SUMO2 and SUMO3 were generated using site-directed mutagenesis. The histone H2B:GFP expression construct (pBOS-H2BGFP) was from BD Biosciences. 

### 4.3. Protein Work

#### 4.3.1. Western Blot Analysis

Cells were harvested by trypsinisation, washed and snap frozen. Cell pellets were lysed in CelLytic M (Sigma) containing protease-inhibitor cocktail (Roche Complete Mini) according to the manufacturer’s recommendations. When assaying SUMOylation the isopeptidase inhibitor N-ethylmaleimide (Sigma) was included at 20 mM final concentration. 

Lysates were cleared by centrifugation at 13,000 rpm and fractionated on SDS-PAGE gels (XCell SureLock Mini-Cell system, Invitrogen). For routine TOP2A resolution 6% or 8% Tris-Glycine SDS-PAGE gels were used and transferred in Tris-Glycine buffer containing 20% methanol, without SDS. For quantification of levels of SUMOylated TOP2A CTR, lysates were resolved using NuPAGE Novex Tris-Acetate 3–8% gradient mini gels and transferred using NuPAGE Transfer buffer (Invitrogen). 

For fluorescence immunoblotting PVDF-FL membrane was used with the blocking and primary antibody steps performed in Odyssey blocking buffer (LI-COR Biosciences, Lincoln, NE, USA): Tris-buffered saline with 0.1% Tween-20 (TBST) (1:1). Secondary antibody incubation was in TBST plus 5% powdered milk. Fluorescence intensities were determined using a CCD scanner (Odyssey; LI-COR Biosciences) according to the manufacturer’s instructions using Image Studio software. For chemiluminescent detection PVDF-P membrane was used, with all steps being performed in TBST plus 5% powdered milk. Antibody-antigen complexes were detected using ECL Plus or ECL prime according to the manufacturer’s instructions (GE Healthcare, Little Chalfont, UK).

#### 4.3.2. Immunoprecipitation

Immunoprecipitation of GFP fusion proteins was undertaken using the GFP-Trap M system (Chromotek, Planegg-Martinsried, Germany).

### 4.4. Microscopy

Fixed cells were viewed using a Leica DM6000B upright fluorescence microscope and images captured using the Leica application suite (LAS) AF6000 system. Where required, cells were imaged in a Z stack (ca. 100–130 steps) and processed using non-blind 3D deconvolution.

#### 4.4.1. Assessment of Mitosis and Nuclear Phenotypes

Cells were plated onto SuperFrost (VWR) slides in chambers 24 h. before processing. Cells were fixed in −20 °C 100% methanol for 13 min., air-dried and mounted using DAPI (0.5 μg mL^−1^) in Vectashield (Vector Labs, Peterborough, UK). DNA was observed under a 40× air objective and phenotypes scored by eye. 

#### 4.4.2. Indirect Immunofluorescence

Cells were grown overnight on SuperFrost (VWR) slides and fixed in PTEMF for 10 min (20 mM Pipes pH 6.8, 0.2% Triton X-100, 10 mM EGTA, 1 mM MgCl_2_, 4% paraformaldehyde). Blocking (10% foetal bovine serum), antibody dilutions and washes were all undertaken in PBS Tween 20 (0.05%). DNA was counterstained with DAPI (0.5 μg mL^−1^) and mounted in Vectashield (Vector Labs). 

#### 4.4.3. Direct GFP Visualisation

To visualise GFP-tagged TOP2A directly, cells were fixed in −20 °C 100% methanol, DNA counter-stained and mounted as described above. GFP was observed under a 100× oil immersion objective and localisation phenotypes scored by eye, within 36 h of fixation. 

#### 4.4.4. Fluorescence Recovery after Photobleaching (FRAP)

Cells were grown on a 8-well chambered coverglass (Lab-Tek) for 24–48 h. with the medium changed to phenol red-free immediately prior to imaging. The temperature of the stage and incubation chamber were maintained at 37 °C throughout. FRAP experiments were performed on a Zeiss LSM 540 confocal microscope. All imaging was carried out using a 100× oil immersion objective and the 488 nm laser line set at 3–5%. Photobleaching was performed using the 405 nm laser at 100% output, scan speed 2, for a single iteration. The size of the bleached area was a 15 × 15 pixel square. Each image was taken at 8× digital zoom, the image size was 400 × 400 pixels and scanning was set to bidirectional with 12 bit data depth and scan speed 9. The pinhole was set at 1.34 Airy Units to produce an optical slice of 1 μm. Three Z-slices were acquired for each timepoint covering 2.0 μm. Imaging time was ca. 1.6 s. per frame; time between frames was 2.0 s; 5 frames were captured pre-bleaching and the area was followed for 4 or 6 min post-bleaching. Movies were processed using ImageJ software (https://imagej.net/ImageJ). Where chromosomes had moved extensively within the field of view, frames were re-aligned using the StackReg plugin (http://bigwww.epfl.ch/thevenaz/stackreg/). The bleached roi was followed throughout the movie by selecting the position in regular slices and completing the remaining positions using the Interpolate ROIs function. Areas measured were: (i) the bleached region (bleach); (ii) an area of the same size lacking fluorescent structures (background); and (iii) the full view imaged. The data was normalised against the background and reference, as well as the pre-bleach signal intensity. Curve fitting and comparison of the recovery curve and t_1/2_ were performed as described [69]. An *F*-test was used to choose the equation and compare datasets. The recovery was fitted to a bi-exponential equation. 

### 4.5. Cell culture

#### 4.5.1. Cell Lines 

Hek-293T and HTETOP cells were grown routinely in Dulbecco’s modified Eagle’s medium containing glutamax, 10% foetal bovine serum, penicillin, streptomycin (all from Invitrogen-Gibco, Waltham, MA, USA) at 37 °C. HTETOP is a HT1080-derived conditional null mutant for TOP2A [43] in which transcription of the Tet-regulatable TOP2A transgene can be repressed by growth in medium containing 1 μg mL^−1^ doxycycline (Sigma). Both TOP2A alleles have been disrupted and the cells retrofitted with an exogenous TOP2A cDNA (referred to here as untagged TOP2A) that is controlled by a Tet transactivator (tTA). 

#### 4.5.2. Stable Transfections

For stable transfection of HTETOP, cells were electroporated using standard conditions: 6 × 10^6^ cells in 800 μL Dulbecco’s Phosphate-Buffered Saline (D.PBS(A)) using a BioRad Pulser II with 30 μg of linearized plasmid DNA at 250 μF, 400 V and 200 Ω. After 24–48 h, puromycin (Merck) selection was applied at 0.5 μg mL^−1^.

#### 4.5.3. Transient Transfections

Transient transfections were undertaken in 6-well dishes according to the manufacturer’s instructions using either Lipofectamine 2000 (Invitrogen) (HTETOP) or TransIT-293 transfection reagent (Mirus) (Hek-293T). Cells were processed 24–48 h post-transfection.

#### 4.5.4. Drug Treatments

The reversible CDK1 inhibitor RO-3306 (Calbiochem, supplied through Merck, Darmstadt, Germany) was resuspended in dimethyl sulphoxide (DMSO) and added overnight (18 h) at a 5 μM final concentration.

### 4.6. Statistical Analyses

Ordinary one-way ANOVA analyses with the Holm-Sidak’s multiple comparisons test (Figure 2) were undertaken within GraphPad Prism 8. Student’s *t*-test and Ordinary one-way ANOVA analyses with post-hoc Tukey HSD test for multiple comparisons (Figure 3 and Figure 4) were undertaken using the online calculator at: http://astatsa.com/. All statistical analysis of FRAP data (Figure 5) was done within GraphPad Prism 8 as described [69].

## Figures and Tables

**Figure 1 ijms-20-01238-f001:**
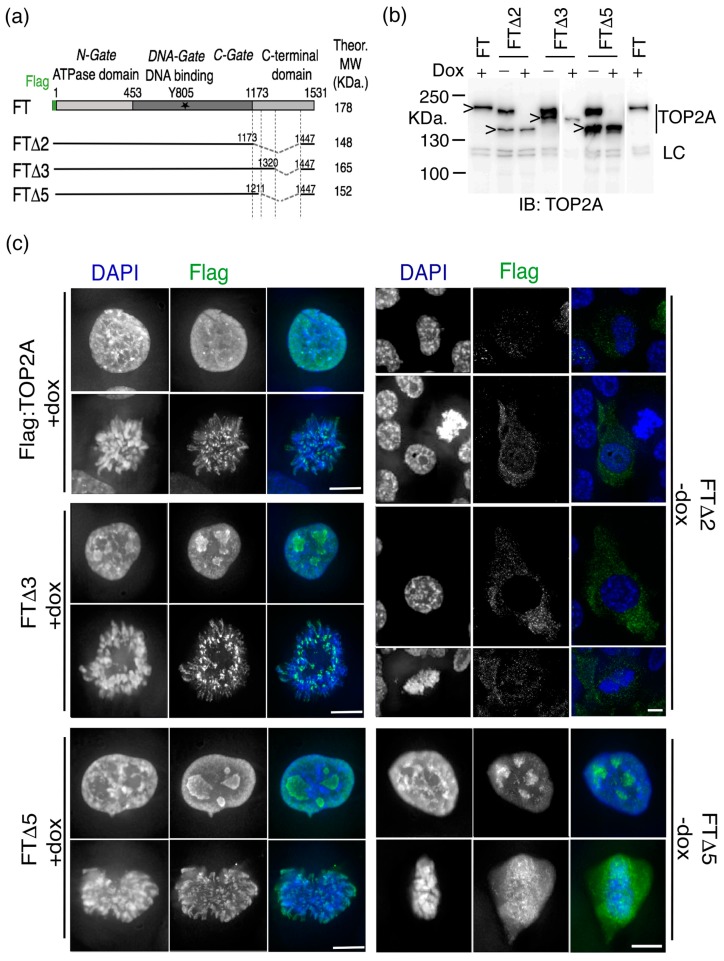
The impact of internal deletions of the CTD on the mitotic localisation of TOP2A (**a**) Schematic of human TOP2A showing the domain structure: the N-terminal ATPase gate (consisting of the ATPase and transducer domains); the DNA-binding gate (consisting of the TOPRIM domain, the Winged Helix Domain (with the active site tyrosine 805) and the Tower domain); the C-gate (formed by the coiled-coil domain); and the unstructured C-Terminal Domain (CTD). Shown below are the internally deleted variants analysed. In each the terminal amino acids 1447–1531, which encompass the main nuclear localisation signal (NLS) and the chromatin tether domain (ChT) are retained. (**b**) Western blotting of whole cell lysates from HTETOP-derived transfectants stably expressing Flag-tagged TOP2A, either full length (FT) or internally deleted variants (FTΔ2, Δ3 and Δ5). The antigen recognised by the TOP2A isoform-specific antibody is retained in all variants. Transfectants have been grown in the presence or absence, of doxycycline. The full length untagged TOP2A protein (~170 KDa.) present in the parental HTETOP cells is transcribed under a tet-regulatable promoter and is suppressed by doxycycline, while the Flag-tagged versions of the protein (expressed under a CMV promoter) are unaffected (indicated by “>”). LC = an unidentified protein detected by the TOP2A antibody and described previously [45]. (**c**) Immunofluorescence localisation of the internally deleted forms of TOP2A in interphase and mitotic cells of stable transfectants, detected using an anti-Flag antibody (green), shown alongside cells expressing the Flag-tagged full length protein (FT). DNA has been counterstained with DAPI (blue). FT (Flag:TOP2A WT) and FTΔ3 cells are shown grown in the presence of dox (the localisation phenotypes were unchanged in the absence of dox/presence of untagged TOP2A WT protein). The FTΔ2 cells shown had been passaged in the absence of dox (long term growth in doxycycline is lethal). FTΔ5 cells grown in both the presence or absence, of dox are shown. This variant localises normally to the nucleus but its targeting of mitotic chromatin is perturbed. This disruption is especially severe in the presence of full length TOP2A protein, where the Flag-tagged variant appears largely cytosolic. Scale bar, 10 μm.

**Figure 2 ijms-20-01238-f002:**
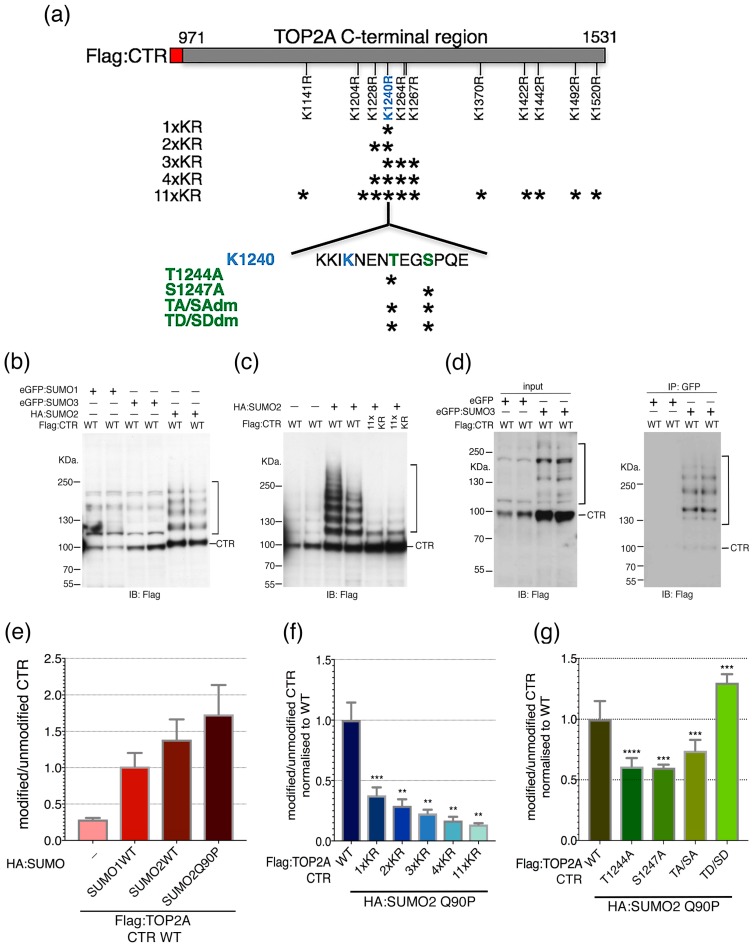
Assaying the biological importance of potential SUMO acceptor sites within the TOP2A C-terminal region (CTR) by Hek-293T transient transfection. (**a**) Schematic of residues tested within the TOP2A CTR. (**b**,**c**) Representative immunoblots probed with anti-Flag antibody. Whole cell extracts were prepared 24–48 h post-transfection and resolved on 3–8% Tris-Acetate gradient gels. (**b**) Cells were co-transfected with Flag:CTR and either eGFP:SUMO1/3 or HA:SUMO2. (**c**) Cells were co-transfected with HA:SUMO2 and either WT or 11xKR-substituted CTR. In (**b**,**c**) samples in paired lanes (e.g., lanes 1 & 2, 3 & 4, 5 & 6) were from independently transfected wells from the same experiment. (**d**) eGFP:SUMO (or eGFP alone) and Flag:CTR co-transfected samples were subjected to GFP immunoprecipitation and detected using anti-Flag antibody. Input cell lysates were resolved on an 8% Tris-Glycine SDS-PAGE gel and IP samples on a 3–8% Tris-Acetate gradient gel. The samples in paired lanes are duplicates. (**e**) Graph showing levels of HMW Flag:CTR products relative to the unmodified ~100 KDa. Flag:CTR in cells expressing endogenous SUMO only or various forms of HA:SUMO (SUMO1 WT, SUMO2 WT or SUMO2 Q90P). For each transient transfection multiple independent experiments were undertaken (*n*, 3–6). Errors bars represent s.e.m. (**f**) Effect of selected K/R substitutions on CTR SUMOylation levels following co-expression with HA:SUMO2 Q90P. Levels were assessed by fluorescence immunoblotting and quantitation of the HMW Flag:CTR ladder relative to the unmodified ~100 KDa CTR. The value for each K/R-substituted CTR was normalised against the mean WT CTR value (n8, un-matched). For each variant, multiple independent experiments were undertaken (*n*, 2–8), with multiple replicates of each sample analysed. (**g**) Effect of selected phosphosite mutations on CTR SUMOylation following co-expression with HA:SUMO2 Q90P. Levels were estimated by fluorescence immunoblotting and quantitation of the HMW Flag:CTR ladder relative to the unmodified ~100 KDa CTR. The value for each CTR variant was normalised against the WT CTR value from within the same experiment (matched). For each variant, multiple independent experiments were undertaken (*n*, 4–9), with multiple replicates of each sample analysed. In (**f**,**g**) Error bars represent s.e.m. Significance was calculated by performing an ANOVA analysis to compare all samples to WT with the P value corrected using the Holm-Sidak’s multiple comparisons test. **** *p* < 0.0001, *** *p* = 0.0001–0.001, ** *p* = 0.001–0.01.

**Figure 3 ijms-20-01238-f003:**
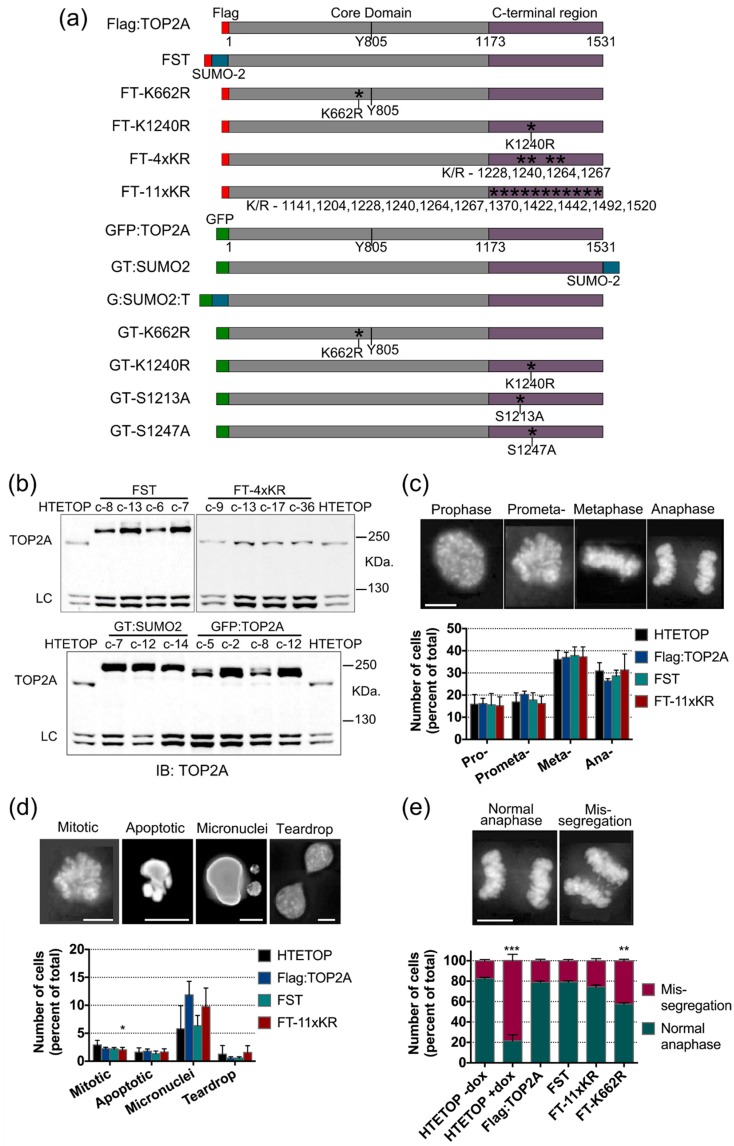
Impact of TOP2A variants on mitotic progression and chromosome segregation. (**a**) Schematic showing the Flag- (red) and GFP (green)-tagged TOP2A variants studied in this work. (**b**) Representative immunoblots from HTETOP-derived transfectants stably expressing variant forms of TOP2A. Transfectants have been grown in the presence of doxycycline to repress the full length untagged TOP2A protein (~170 KDa) present in parental HTETOP cells. LC = an unidentified protein detected by the TOP2A antibody and described previously [45]. (**c**,**d**) Representative images and percentage of cells found at each stage of mitosis (**c**) and showing various nuclear phenotypes (**d**). Cells were scored by eye at 40× magnification. Three transfectant clones were tested for each variant and three independent experiments were performed. Approx. 500 mitotic cells and 1000 nuclei were scored per repeat. Slides were fixed in methanol and DNA stained with DAPI. Error bars represent standard error of the mean (s.e.m.). Significance was assessed by performing a one-way ANOVA analysis and post-hoc Tukey HSD test to compare lines to the HTETOP (-dox) control. (**e**) Representative images and percentage of cells, showing normal chromosome segregation and mis-segregation in anaphase. Cells were fixed in methanol and DNA stained with DAPI. Scale bar: 10 μm. Growth of the conditional null mutant line HTETOP in doxycycline for 48 h results in high levels of anaphase mis-segregation. Expressing Flag-tagged TOP2A in dox-treated HTETOP cells rescues the mis-segregation phenotype. Three transfectant clones were tested for each Flag-tagged variant. Three independent experiments performed with approx. 300 cells scored for each repeat. Error bars represent s.e.m. Significance was assessed by performing a one-way ANOVA to compare lines expressing the various Flag-tagged TOP2A variants to the HTETOP-dox control with the *p* value corrected using the post-hoc Tukey HSD analysis for multiple comparisons. *** *p* = 0.0001–0.001, ** *p* = 0.001–0.01, * *p* = 0.01–0.05.

**Figure 4 ijms-20-01238-f004:**
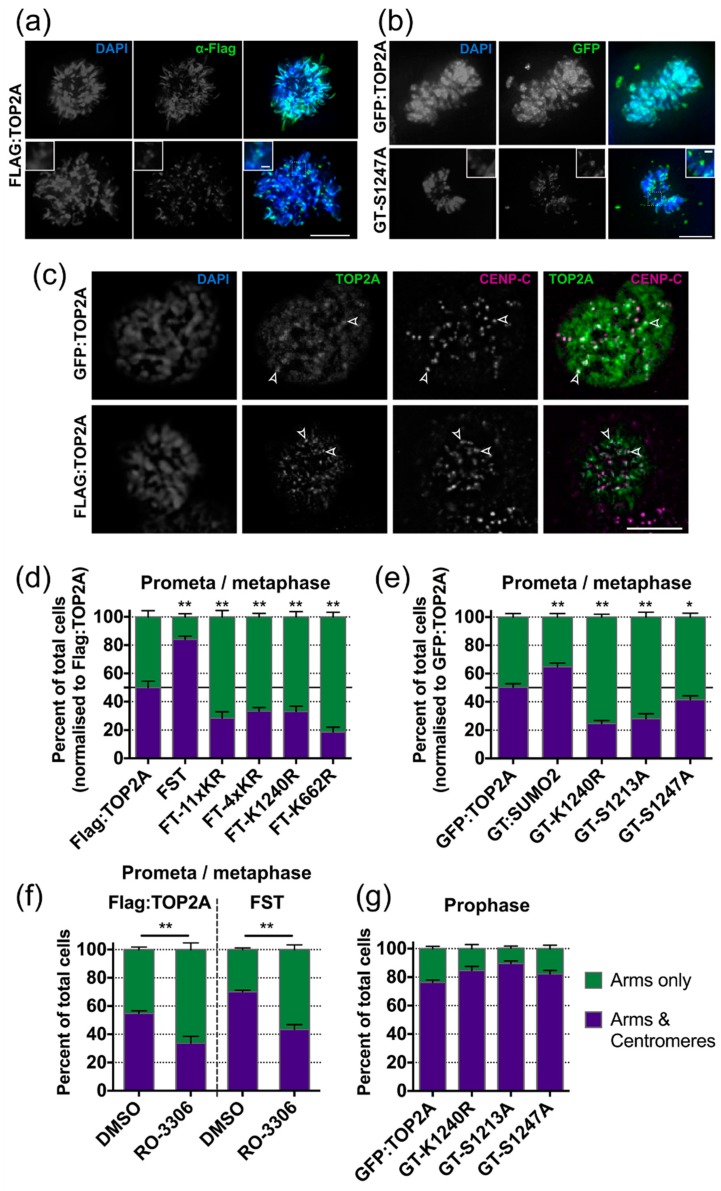
TOP2A mitotic localisation phenotypes. (**a**) Representative images of the main TOP2A localisation patterns displayed in mitotic cells expressing Flag-tagged TOP2A (unmutated). Cells were fixed in formaldehyde (PTEMF) and stained using anti-Flag antibody (green). DNA was stained using DAPI (blue). Two main phenotypes were observed: staining along the chromosome arms only or on the arms together with more intense accumulation at centromeres. (**b**) Representative images of the main TOP2A localisation patterns displayed in mitotic cells expressing GFP-tagged TOP2A (unmutated and S1247A). Cells were fixed in methanol and GFP visualised directly (green). DNA was stained using DAPI (blue). Scale bars in (**a**,**b**): main image, 10 μm; inset 1 μm. (**c**) Immunofluorescence images showing co-localisation of punctateTOP2A signals with centromeres. Cells were fixed in formaldehyde (PTEMF) and stained using anti-TOP2A antibody (green) and anti-CENP-C antibody (magenta). DNA was stained using DAPI. The open arrowheads indicate examples of TOP2A and CENP-C co-localisation. Scale bar: 10 μm. (**d**) Graph of fixed prometaphase and metaphase cells showing the main TOP2A localisation patterns in stable cell lines expressing various Flag-tagged TOP2A forms. The value for each variant was normalised against the Flag:TOP2A (unmutated) from within the same experiment, with the WT value set as 50%. Three clones were analysed for each TOP2A variant, with three repeats performed for each clone and ~200 cells scored per repeat. Significance was estimated by one-way ANOVA to compare all lines to the Flag:TOP2A (unmutated) with the *p* value corrected using the post-hoc Tukey HSD analysis for multiple comparisons. Error bars represent s.e.m. (**e**) Graph of fixed prometaphase and metaphase cells showing the main TOP2A localisation patterns in stable cell lines expressing various forms of GFP-tagged TOP2A. Data collection and statistical analysis were performed as described for (**d**). (**f**) Graph showing the percentage of fixed prometaphase and metaphase cells showing the main TOP2A localisation patterns in Flag:TOP2A and FST cell lines one hour after release from overnight (18 h) treatment with the CDK1 inhibitor RO-3306 (or 0.1% DMSO alone). Two independent experiments (each consisting of two parallel replicates) were performed and ≥100 cells scored per sample. Error bars represent s.e.m. Significance was assessed by performing a Student’s *t*-test. (**g**) Graph showing the percentage of fixed prophase cells showing intense punctate centromeric TOP2A signals in stable cells lines expressing various GFP-tagged TOP2A forms. Prophase cells were identified on the basis of cyclin B1 staining. One clone was analysed for each TOP2A variant, with three repeats performed for each variant and ~100 cells scored per repeat. Error bars represent s.e.m. ** *p* = 0.001–0.01, * *p* = 0.01–0.05.

**Figure 5 ijms-20-01238-f005:**
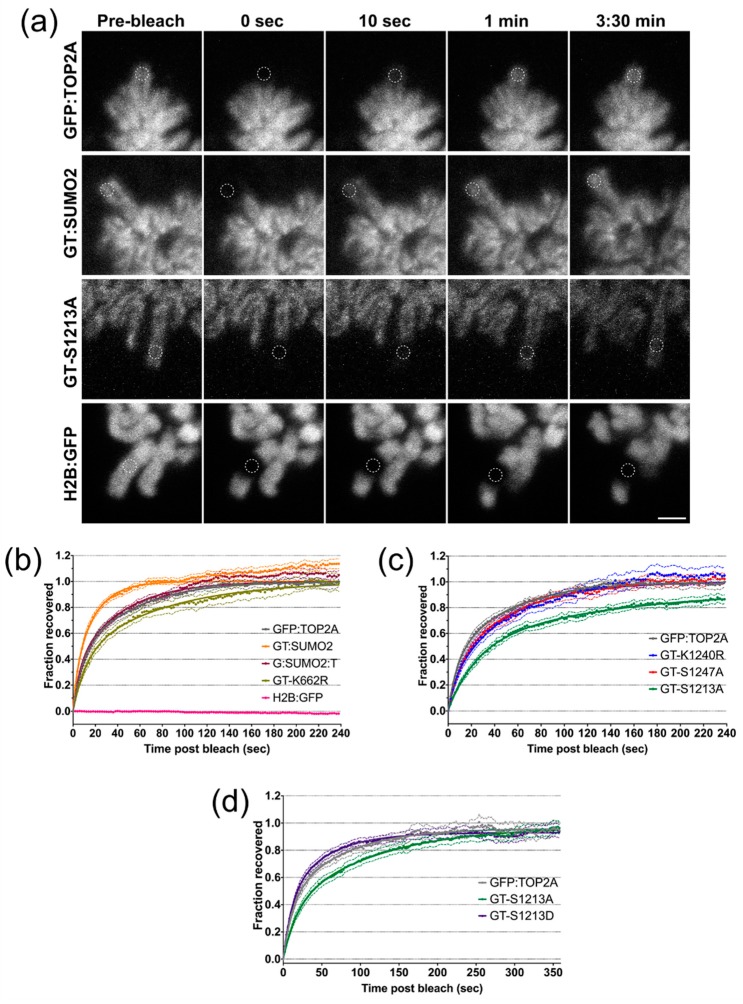
The mobility of TOP2A on mitotic chromatin. (**a**) Still images from FRAP movies showing the last frame before bleaching (Pre-bleach), the first frame post-bleach (0 s) and three frames showing progress of recovery at the indicated timepoints. The white dashed line indicates the circular region that has been bleached. Representative images shown for the GFP:TOP2A, GT:SUMO2 and GT-S1213A cell lines respectively. Also included is a cell line expressing H2B:GFP. Scale bar: 2 μm. (**b**–**d**) FRAP recovery plots. Normalised mean recovery for the cell lines indicated. Three independent replicates were performed per cell line, with at least 5 movies analysed per replicate. Error bars representing s.e.m. are shown around each recovery curve.

**Table 1 ijms-20-01238-t001:** Summary of the impact of CTD deletions on Flag:TOP2A localisation and ability to complement dox-induced lethality in stably transfected HTETOP cells.

Expression Construct	Residues Deleted	Complementation (Growth in Dox)	TOP2A Localisation *	
			Interphase	M Phase
FT (Flag:TOP2A)	none	Y	Nuc	Arm + cen
FTΔ2	1173–1446	N	N + C	diffuse
FTΔ3	1321–1446	Y	Nuc	Arm + cen
FTΔ5	1212–1446	Y	Nuc	diffuse

Complementation was assessed by long term cell growth in doxycycline, which represses expression of the untagged full length TOP2A transgene. * the localisation pattern was found to be the same irrespective of the presence/absence of full length TOP2A (−/+ dox) except for the FTΔ5 variant, whose diffuse localisation to mitotic chromatin was even more severely disrupted by the presence of full length TOP2A. Nuc, nuclear; N + C, nuclear & cytoplasmic; arm + cen, localisation to mitotic chromatin with frequent axial arm and punctate centromere staining; diffuse, throughout mitotic cell with little/no accumulation on mitotic chromatin.

**Table 2 ijms-20-01238-t002:** Analysis of FRAP recovery curves (time period: 240 s).

Cell Line	Fast t_1/2_ (s) *	Slow t_1/2_ (s) *	Immobile Fraction	Curve Fitting Compared to GFP:TOP2A	
				Goodness of Fit (R Square)	Significance (*p* Value)
GFP:TOP2A	5.15 ± 1.13	33.49 ± 2.86	6.71 × 10^−11^	0.997	NS (>0.9999)
GT-K662R	8.12 ± 2.59	52.98 ± 11.76	1.14 × 10^−16^	0.892	<0.0001
GT:SUMO2	2.59 ± 1.24	12.82 ± 1.20	1.33 × 10^−16^	0.530	<0.0001
G:SUMO2:T	2.53 ± 1.05	23.87 ± 1.24	1.25 × 10^−16^	0.947	<0.0001
GT-K1240R	2.87 ± 2.30	31.47 ± 2.64	1.72 × 10^−16^	0.945	NS (0.4219)
GT-S1247A	5.16 ± 1.91	34.03 ± 3.00	1.22 × 10^−16^	0.974	NS (0.8093)
GT-S1213A	16.17 ± 4.11	85.14 ± 43.43	0.066	0.186	<0.0001

The recovery has been fitted to a biexponential equation as described in Reference [69]. R^2^ was calculated using the mean curve for each cell line and *p* value calculated using a replicates test. * = mean value ± standard error of the mean.

**Table 3 ijms-20-01238-t003:** Analysis of FRAP extended recovery curves (time period: 360 s).

Cell Line	Fast t_1/2_ (s) *	Slow t_1/2_ (s) *	Immobile Fraction	Curve Fitting Compared to GFP:TOP2A	
				Goodness of Fit (R Square)	Significance (*p* Value)
GFP:TOP2A	9.4 ± 2.1	61.15 ± 14.96	0.015	0.968	NS (0.9632)
GT-S1213A	10.47 ± 2.94	71.87 ± 15.09	0.035	0.653	<0.0001
GT-S1213D	9.31 ± 1.69	45.45 ± 9.73	0.055	0.976	NS (0.7924)

The recovery has been fitted to a biexponential equation as described in Reference [69]. R^2^ was calculated using the mean curve for each cell line and *p* value calculated using a replicates test. * mean value ± standard error of the mean.

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
