# Peer review of "The Impact of the C-Terminal Region on the Interaction of Topoisomerase II Alpha with Mitotic Chromatin"

_ijms, 2019, doi:10.3390/ijms20051238_

Round 1
Reviewer 1 Report
This paper is valuable because it closes a gap. It is long known that the two isoforms of mammalian Top 2 serve distinct biological functions. These are encoded by the C-terminal domains of the enzymes, which are dispensable for catalytic activity of the enzymes. Moreover it has long been realised that the alpha-isoform serves unique functions in mitosis and chromosome condensation that are not supported by the beta-isoform. However, so far, it has not been sorted out, precisely which features (single amino acid residues and modifications thereof) within the alpha-carboxyterminal domain are essential for directing the enzyme to it's specific and exclusive areas of activity within the mitotic chromatin. The present paper provides exactly that information.
The authors use a cellular complementation assay (HTETOP) previously established to demonstrate the biological role of the Top2-CTDs in general to test a series of internally deleted forms and point mutations of Top2a thereby identifying a set of small sequence motifs within the alpha-CTD that contribute to the mitotic localisation of Top2alpha. Next they address sumoylation within these sequences by testing a series of candidate lysin-replacements in a transient transfection assay. The thus identify K1240 as a major SUMO-acceptor site and they show by MassSpec that sumoylation at that side is affected (regulated) by phosphorylation at serine and threonine residues nearby which are known to be hyper-phosphorylated during mitosis. Next they use the above complementation assay to test the relevance of SUMO-/phosphorylation around K1240 and also at K662 (another known sumoylation site) for the functioning of Top2A during mitosis. They show that lysine replacements at these sites as well as serine replacements at the phosphorylation sites near K1240 alter localisation of TOP2 on chromosomes and disrupt mitotic progression and chromosome segregation. For sumoylation at lysine 662 they also show that it plays a role in the exchange-rate of the protein at its sites of activity on mitotic chromatin. This may be one of the mechanism by which sumoylation regulates net-occupation of mitotic chromatin by Top 2A, although the authors refrain from elaborating this point adequately.
The paper comprises an impressive set of experimental data, which support the main conclusions beyond any doubt. I cannot think of any experiment or control that would be missing. The information thus obtained is interesting and completes our understanding of how the essential support of mitosis by Top2A works in detail. This information definitely merits rapid publication.
The paper is well written. The introduction provides an adequate overview of the point at which the study starts. The results are well presented an adequately described. The weakest part is the discussion, which fails to disengage from the technical details and to provide a holistic and concise synopsis of what has been found out. It would be a pity to publish such a nice set of data without a proper take-home message.
Hence my only suggestion for improvement: The authors could be bolder. I am missing a conclusive sentence at the end of each results section describing what the respective set of experiments has brought to light. I am missing a brief initial paragraph in the introduction that summarises the salient findings. I am missing a brief final paragraph in the discussion that provides a condensed and comprehensive view on the TOP2A-targeting-mechanism in the light of the new data (i.e. the take-home message).
Otherwise this is a very nice piece of work.
Author Response
We thank the referee for their very positive comments.
In response to the suggestion that we could be bolder we have added a section to the end of the introduction (briefly outlining the main findings) (lines 80-83) and to the end of the discussion (with the take-home message) (lines 593-601).
Reviewer 2 Report
In the manuscript, “The impact of the C-terminal region on the interaction of topoisomerase II alpha with mitotic chromatin”, Antoniou-Kourounioti et al., using stable cell lines to study DNA truncation, IFA, mass spectrum and Fluorescence recovery after photobleaching (FRAP) to study the functional molecular mechanisms about the chemical modification of CTD domain of TOP2A. Their results suggest that SUMOylation and phosphorylation of specific residues within the TOP2A CTD can regulate the enzyme’s residence time on mitotic chromatin. Furthermore, these effects can influence the protein’s subcellular localization, faster exchange between molecules in the chromatin and apparently facilitating centromere targeting for cytosolic pools of TOP2A.
These studies were conducted very well and the data were solid to support their conclusions. And, the data presentation and writing are very clear and easy to follow.
Some minor comments are listed below:
1. Line 86-87: Subsequently Clarke and colleagues demonstrated that 86 the most distal 30 amino acids (residues 1500-1531),…….
Residues between 1500 to 1531 should contain 32 amino acids instead 30 amino acids.
2. Line 69-70: For example, work using Xenopus 69 egg extracts (XEE) has shown that Top2a is a major SUMOylation target during mitosis,……
Top2a should change to Top2A?
3. Line 591-593: SUMOylation, or phosphorylation, of mitotic TOP2A can change the protein’s pI, possibly disrupting its interaction with DNA and promoting faster exchange between molecules in the chromatin and cytosolic pools.------
I learned the chemical modification had changed the pI of TOP2A, however, does the modification change the conformation? And, is it possible to discuss the possible the tertiary or quaternary structure change after the modification?
Author Response
We thank the referee for their positive comments.
1. lines 86-87. This has been corrected.
2. lines 69-70. We have adopted “Top2a” for the Xenopus topoisomerase II alpha protein (listed under “Abbreviations”) and hence have left this as it stands.
3. Lines 591-593. Since it has not been possible to crystallise the CTD of eukaryotic TOP2 it is difficult for us to speculate on the possible impact of these PTMs on the protein’s tertiary/quaternary structure. While the CTDs of bacterial type 2 topoisomerases (gyrase and Topo IV) have been shown to form beta-pinwheel structures around which the DNA is wrapped, there is no conservation, at the protein sequence level, between bacterial and eukaryotic type IIA topoisomerases. We have pointed out this deficiency in the current understanding in our concluding paragraph (lines 593-601).